# Growing Nano-SiO_2_ on the Surface of Aramid Fibers Assisted by Supercritical CO_2_ to Enhance the Thermal Stability, Interfacial Shear Strength, and UV Resistance

**DOI:** 10.3390/polym11091397

**Published:** 2019-08-26

**Authors:** Luwei Zhang, Haijuan Kong, Mengmeng Qiao, Xiaoma Ding, Muhuo Yu

**Affiliations:** 1State Key Laboratory for Modification of Chemical Fibers and Polymer Materials, College of Materials Science and Engineering, Donghua University, Shanghai 201620, China; 2School of Materials Engineer, Shanghai University of Engineer Science, Shanghai 201620, China

**Keywords:** aramid fiber, nano-SiO_2_, supercritical CO_2_, thermal stability, interfacial shear strength, UV resistance

## Abstract

Aramid fibers (AFs) with their high Young′s modulus and tenacity are easy to degrade seriously with ultraviolet (UV) radiation that leads to reduction in their performance, causing premature failure and limiting their outdoor end use. Herein, we report a method to synthesize nano-SiO_2_ on AFs surfaces in supercritical carbon dioxide (Sc-CO_2_) to simultaneously improve their UV resistance, thermal stability, and interfacial shear strength (IFSS). The effects of different pressures (10, 12, 14, 16 MPa) on the growth of nanoparticles were investigated. The untreated and modified fibers were characterized by Fourier transform infrared spectroscopy (FT-IR), X-ray diffraction (XRD), X-ray photoelectron spectroscopy (XPS) and scanning electron microscopy (SEM). It was found that the nano-SiO_2_-decorated fibers exhibited improvement of thermal stability and mechanical properties, and the IFSS of the nano-SiO_2_ modified fibers increases by up to 64% compared with the untreated fibers. After exposure to 216 h of UV radiation, the AFs-UV shows a less decrease in tensile strength, elongation to break and tensile modulus, retaining only 73%, 91%, and 85% of the pristine AFs, respectively, while those of AFs-SiO_2_-14MPa-UV retain 91.5%, 98%, and 95.5%. In short, this study presents a green method for growing nano-SiO_2_ on the surface of AFs by Sc-CO_2_ to enhance the thermal stability, IFSS, and UV resistance.

## 1. Introduction

Materials with extraordinary mechanical properties are required in industrial fields. Aramid fiber (AF) is highly praised due to its high strength and high modulus. Regarded as a promising candidate for advanced composite matrixes [1,2,3], AF has been intensively adopted in high-performance and heavy-industry fields, seeing aerospace, automotive, shipbuilding, sport, and military applications [4,5,6]. However, the abundant intermolecular hydrogen bonding and high degree of crystallinity lead to a smooth and inert surface of AFs, which hinders the adhesion strength of AFs to epoxy matrices, limiting its application in composite materials [7]. Moreover, the large amount of amide bonding in fibers is easily decomposed by UV irradiation [8,9], high temperature [10], and moisture [11]. These drawbacks constrain the working durability of AF-based composites under long-term outdoor conditions [12]. 

In order to prevent AFs from losing their mechanical properties in harsh environments, advanced studies have conducted extensive modification methods on their surfaces [13,14,15,16,17]. Among these methods, coating with nano-SiO_2_ has gained much attention due to its attractive features of high absorption of ultraviolet light, transparency in the visible region, a stable mesoporous structure, eco-friendliness, low cost, excellent biocompatibility, and nontoxicity in nature [18,19,20]. As a versatile liquid-based method for synthesizing nanoparticles and oligomers inorganic networks [21], the sol-gel process has shown great advantages in obtaining nano-SiO_2_ from the hydrolysis and condensation of alkoxide precursors [22] and has a wide range of applications in hybrid coating materials [23]. Since the sol-gel method is a simple procedure, it has attracted great interest in the fields of material synthesis and polymeric modification. Worth mentioning is that the AFs could be modified with the mesoporous structure of a silica aerogel prepared by the sol-gel method to make up for the shortcomings of the fiber in application, thus improving the interfacial strength of fiber-reinforced composites [24,25,26]. 

Unfortunately, difficulties were found in controlling the structure, morphology, and composition of the nanoparticles when applying the sol-gel method to the surface modification of fibers [27,28]. With favorable properties, such as being abundant, inexpensive, nonflammable, nontoxic, environmentally benign, and having an adjustable processing temperature and pressure, supercritical carbon dioxide (Sc-CO_2_) has great potential in synthesizing nano- and micromaterials [29]. In recent years, methodologies related to preparing inorganic silica by Sc-CO_2_ have been reported in creative studies where silica aerogel microparticles [30] and hollow silica microspheres (SM) [31,32,33] were synthesized by Sc-CO_2_ with controlled size, distribution, morphology, and composition. However, the fiber modification still did not achieve the desired effect since these synthesized particle sizes were not on the nanoscale. Furthermore, the binding force of the particles to the polymeric surface was not mentioned in these studies. Joabel et al. [34] carried out the deposition of SiO_2_ nanoparticles on cellulose fibers by the sol-gel process, obtaining a hybrid organic–inorganic material through chemical bonding. Arnaud et al. [35] successfully impregnated SiO_2_ into wet coagulated cellulose to form cellulose–silica networks for preparing high-performance composite aerogels. The stable interfacial hydrogen bond structure between nano-SiO_2_ and meta-AFs has been confirmed by Fei et al. [36] through molecular simulation, and the number of hydrogen bonds was associated with the nanoparticle radius. 

The purpose of this study is to grow nano-SiO_2_ on the surface of aramid fibers, assisted by Sc-CO_2_, and to form a stable interface between the inorganic and organic phases by the construction of a nanonetwork structure. During the growing process, hydrogen bonding played a role of the ties and bridges and the capacity of the Sc-CO_2_ to dissolve and diffuse the small molecules, like the precursor, was fully utilized. The shapes and sizes of the nanoparticles were controlled for uniform growth on the aramid surface by using supercritical fluid technology. In this way, the thermal stability, UV resistance, and interfacial shear strength (IFSS) of AFs with resin were enhanced by the decoration of nano-SiO_2_ on the fiber surface. Our results demonstrate the synthesis of nano-SiO_2_ on the surface of AFs, resulting in a material that can meet the requirements of applications in harsh environments.

## 2. Materials and Methods 

### 2.1. Materials

Para-aramid fiber (Afcool, 1200D) was provided by Hebei Silicon Valley Chemical Co., Ltd. (Hebei, China). Ethyl orthosilicate (TEOS) was purchased from Aladdin Industrial Co., Ltd. (Shanghai, China). Ammonia was obtained from Pinghu Chemical Reagent Factory (Zhejiang, China). Polyethylene glycol with a molecular weight of 2000 and trisodium citrate were purchased from Sinopharm Chemical Reagent Co., Ltd. (Shanghai, China). Ethanol (Nuclear, 99.5%) was purchased from Wokai Biotechnology Co., Ltd. (Shanghai, China). Carbon dioxide with a purity of 99% was supplied by White Martins (Shanghai, China). Epoxy resin E44 and curing agent triethylenetetramine were purchased from Kailuo Chemical Co., Ltd. (Guangzhou, China). All these chemicals were of analytical reagent grade and used without further purification. 

### 2.2. Preparation of SiO_2_-Modified Aramid Fibers

Firstly, the AFs were washed with acetone through a sequential soxhlet extraction at 80 °C for 24 h; 2 g of cleaned fibers were hung in a reactor with two-liter volume and peak pressure of 25 MPa. Meanwhile, 22 mL TEOS and 30 mL ethanol were added into the bottom of the reactor. The reaction kettle was closed, the temperature adjusted to 80 °C, and filling with carbon dioxide was conducted to reach the supercritical state (shown in Figure 1a). The TEOS was carried to the surface of AFs with the cosolvents Sc-CO_2_ and ethanol. After maintaining the state for 2 h, a solution containing 30 mL ethanol, 3.6 mL distilled water, 2 mL ammonia, 0.25 g polyethylene glycol, and 0.25 g trisodium citrate was added to trigger the hydrolysis of the TOES. Holding the supercritical state for 2 h, the nanonetwork structure was built on fiber surface by the continuous hydrolysis of the precursor. During hydrolysis, the effects of different supercritical pressures (10, 12, 14, and 16 MPa) on the nano-SiO_2_ growing were investigated. The obtained fibers treated with different pressures were extracted continuously in a supercritical environment three times, ensuring that the residual solvents attached to the fiber surface were removed. The AFs-SiO_2_-10MPa, AFs-SiO_2_-12MPa, AFs-SiO_2_-14MPa, and AFs-SiO_2_-16MPa samples were finally made by drying in an oven at 100 °C for 1 h. A schematic diagram of growing nano-SiO_2_ on the fiber surface is shown in Figure 1b.

### 2.3. Methods of Characterization

Chemical structures of the samples were tested by the KBr disk method with a Thermo Fisher FT-IR spectrometer (NEXUS-670, Thermo Electron Corp, Waltham, MA, USA) in the wavenumber range from 4000 to 400 cm^−1^. 

X-ray diffraction (XRD, D/max-2550VB+/PC, Rigaku Corporation, Japan) was used to determine the crystalline phase of AFs. The measurement was conducted with a radiation source of λ = 1.5406 Å (CuKα, 40 kV, 200 mA), followed by a scanning range of 5.0°–60.0° at a speed of 20 °/min. 

The particle size distribution of nano-SiO_2_ was measured using Nanoparticle size and Zeta potential analyzer, with a particle size ranging from 0.5 nm to 10 µm (Litersizer 500, Beijing, China). 

The surface electronic state of the sample was refracted to X-ray photoelectron spectroscopy (XPS) testing using a Thermo Escalab 250Xi spectrometer (AZoNetwork UK Ltd., Manchester, UK) equipped with an Al anode (AlK = 1486.7 eV). 

The surface morphology of samples was observed using a field-emission scanning electron microscope (FE-SEM, Hitachi Co., Tokyo, Japan). 

The thermal performance of the untreated and nano-SiO_2_-treated aramid fiber was characterized by a TGA analyzer (209 F1, Netzsch, Germany) in a flowing nitrogen atmosphere from room temperature to 900 °C with a heating rate of 20 °C/min. 

UV–Vis measurements of the samples were conducted by the UV–Vis spectrophotometer (UV3600, Shimadzu, Japan) with a wavelength of 250 to 500 nm, and the intensity of the UV light source was 100 µW∙cm^−2^ (365 nm). 

UV radiation tests were done in a UV weathering test machine (40 W, 280~315 nm, lamp length of 1220 mm, Dongguan Instrument Co. Ltd., Guangzhou, China) with a relative humidity of 60% for 216 h (more than 3 months under normal circumstances) at room temperature following the accelerated photoaging procedure according to GB/T 14522-93. 

The measurements of fiber mechanical properties were made on an XQ-1 computerized mechanical tester from Donghua University. The gauge length was 20 mm with a loading speed of 5 mm/min. Twenty samples were tested for each type of sample. 

The surface morphology of the samples before and after UV irradiation was observed by environmental scanning electron microscopy (E-SEM, Hitachi S-4700, Tokyo, Japan). All samples were precoated with a layer of gold in advance and the voltage and current were adjusted at 12.5 kV and 10 µA, respectively, during the test.

### 2.4. Microdebonding Test

In order to evaluate the interfacial adhesion of fibers with epoxy matrix, a microbond technique was employed. The pristine and modified AF used as composite specimens were all selected in a random manner. The epoxy resin system was made of epoxy resin E-44 with tetraamine as a curing agent, which were mixed at the ratio of 10:1 by weight, respectively. A single fiber was mounted on a hard-paper frame and a microdroplet of epoxy (<190 µm) was deposited on each fiber by employing hair “brush” (shown in Figure 2a). Then, the microdroplets were cured in an oven at 120 °C for 5 h. The image of composite specimens shown in Figure 2b was captured by an optical microscope (Nikon Eclipse LV150) and the embedded length and diameter of fibers were measured with optical software. The single fiber pull-out test was done with an Electronic Microsphere Experimental Debonding Tester (YG163, Wenzhou, China). The fiber with the microdroplet was placed on the debonding device (shown in Figure 2c) and a force was applied to the free of the fiber to pull it off the matrix while the force was continuously recorded. The specimen was tested at a constant speed of 10 mm/min and the max force during the pull-out test was recorded for further data analysis. The interfacial shear strength (IFSS) was calculated by Equation (1) [37]:(1)$$\mathrm{IFSS}=\frac{F}{A}=\frac{F}{\pi *D*L}$$
where *F* is the maximum force measured, *D* is the diameter of the fiber, and *L* is the embedded length of the droplet. Each sample was tested for 15 specimens and the values were averaged.

## 3. Result and Discussion

### 3.1. Characterization of Aramid Fibers

Figure 3 gives the FT-IR spectra of the AFs before and after treatment with nano-SiO_2_. The N–H stretching band at 3435 cm^−1^ and the stretching vibration of the amide C=O group at 1652 cm^−1^ are the characteristic peaks unique to AFs. Compared to the spectrum of the untreated AF, the modified AFs display two new adsorption peaks around 1089 and 800 cm^−1^ caused by the symmetric and asymmetric stretching vibrations of Si–O–Si bonds, respectively [38,39,40]. The oxygen atoms attributed to the silica in these bands are set up to bridge between each two silicon sites. Additionally, the peak at 470 cm^−1^, belonging to the bending vibration of Si–O bonds, constitutes the characteristic peak of silica together with the former two peaks [41], which strongly suggests that that nano-SiO_2_ is successfully coated on the AF surface after modification. 

Figure 4a shows the crystal structure of the untreated and nano-SiO_2_-modified AFs. As can be seen, the shape and positions of diffraction peaks are unchanged between the untreated and the modified fibers, indicating that there are no changes in the crystal structure during the modification process. It is worth noting that the intensity of diffraction peaks of the modified fibers are all lower than that of the untreated fibers, which is related to the covering amorphous silica on the surfaces of fibers [42]. In other words, AFs-SiO_2_-14MPa possess the lowest crystallinity in all modified fibers, indicating that the SiO_2_ particles are the richest.

The particle size distributions of SiO_2_ treated under the pressure of 14 MPa are given in Figure 4b. According to the size distribution map, the particle sizes concentrate between 10 and 100 nanometers with an average size of 53 nm, illustrating that the synthesis of SiO_2_ in supercritical CO_2_ (Sc-CO_2_) has reached the nanometer size. 

### 3.2. Surface Chemical Composition of AFs

To confirm the above statements, XPS analysis is used to show the different elemental contents of the samples. As shown in Figure 5a, the C_1s_, O_1s_, and N_1s_ peaks appear at 285, 400, and 532 eV in the wide scan, and they are ascribed to C, O, and N elements in AF, respectively. Compared with the scan of the untreated AFs, the modified AFs present new peaks at 101 and 150 eV, which means that the silica has been successfully grown on the surface of AFs by Sc-CO_2_ [43]. The Si_2p_ core-level spectra of the modified AFs shown in Figure 5b describes the chemical peak of Si element with different intensities, proving the existence of SiO_2_. The differences are further illustrated by the change in the content of the elements in Table 1. It is found that the C and N element concentrations of the untreated fibers are 77.49% and 8.78%, respectively, while those have decreased in the nano-SiO_2_-treated fibers. The O element concentrations of the treated fibers under supercritical pressure of 10, 12, 14, 16 MPa increases to 22.22%, 29.91%, 32.64%, and 20.04%, respectively, and the ratios of O/C of those fibers increase to 0.33, 0.55, 0.65, and 0.29, respectively, compared with the O element concentration of 13.72% and O/C ratio of 0.18 of the untreated fibers. Simultaneously, the Si element concentration of the above-modified fibers increases by 5.1%, 10.68%, 11.69%, and 4.15%, respectively, and the ratios of Si/C of those fibers increase by 0.07, 0.20, 0.23, and 0.06, respectively. The results show that the SiO_2_ contents obtained under pressure of 14 MPa are obviously higher, and the different concentrations of O and Si elements indicate that the growth of SiO_2_ on the surface of AFs has been influenced by the supercritical pressure. Since the concentrations of CO_2_ increase as the pressures increase, the dissolution amounts and diffusion rate of the precursor in Sc-CO_2_ increase, thus contributing to the packed growth of SiO_2_ on the surface of AFs. During the process, the addition of surfactants will generally prevent particle aggregation. Nevertheless, the increased system pressure can cause an increase in static charge on the surface of AFs and particles because of the increase in charged ions with the reaction between CO_2_ and other solvents, eventually causing the increase of charge interaction. When the CO_2_ is released, the fast relative motion between the particles and the gas will induce particle aggregation [44,45]. Since the large particles are easily detached under the external forces, low coating of nano-SiO_2_ at 16 MPa results. That means that only when reaching an appropriate supercritical pressure can the SiO_2_ grow adequately on the surface of AFs. 

### 3.3. Surface Morphology of the Aramid Fibers

The surface morphology of the fibers was observed by FE-SEM technology as shown in Figure 6. The untreated AF surface shown in Figure 6a is clean and tidy, and the nanoscale silica spheres with different morphologies and sizes can be observed on the modified AFs as shown in Figure 6b–i. Compared with the spherical nano-SiO_2_ obtained under the pressure of 10 MPa shown in Figure 6b, the nano-SiO_2_ formed under 16 MPa (Figure 6h,i) owns a unique shape like a “snowflake” or a “leaf” and has smaller particles and rarer numbers than that under 10 MPa. Surprisingly, the nano-SiO_2_ particles synthesized under the pressure of 12 MPa show a tendency to decrease from large sizes, compared with the particles produced under the pressure of 10 MPa. When the prepared nano-SiO_2_ is under the pressure of 14 MPa, the particles exhibit a more numerous and densely packed morphology. The tendency of the particle size to become smaller gradually is attributed to the increase of the pressure, leading to the increased diffusion rate of molecules, which makes SiO_2_ particles grow quickly without accumulating into large particles [29]. However, as discussed in the XPS analysis, the excessive pressure will make the particle sizes larger due to the aggregation, which is attributed to the formation of leaf-shaped particles under 16 MPa. Moreover, a honeycomb-shaped nano-SiO_2_ is observed on the surface of AFs treated at 14 MPa, which will protect the fibers from environmental erosion. Furthermore, the roughness of the fiber surface is improved, which is beneficial to improve the interfacial strength of the fiber/epoxy composites. 

### 3.4. Thermal Properties of the Aramid Fibers

Thermal stability of the untreated AFs and the modified AFs was typically evaluated using TGA. As shown in Figure 7, all samples have two thermal decomposition stages during the heating process. The first stage at 100–180 °C is attributed to the evaporation of absorbed water and the second stage at 520–610 °C is caused by the severe degradation reactions, cross-linking, or carbonization reactions [14]. However, the quality of nano-SiO_2_-modified AF is higher than that of unmodified AF even at the same temperature and has a higher residual quality shown in Table 2. Meanwhile, the residual quantity of modified AF treated under pressure of 14 MPa increased by 4.16% than the untreated AF. It can be explained from this aspect: silica can migrate to the molten polymer surface during degradation, forming a barrier that physically protects the remaining polymer from heat [46]. Since the coating of SiO_2_ treated under 16 MPa become less, the residual mass correspondingly reduces. The higher the contents of SiO_2_ nanoparticles, the better the thermal properties of AFs.

### 3.5. UV Stability

#### 3.5.1. UV–Vis

Figure 8a shows the UV–Vis spectra of AFs treated with different pressures in the wavelength range from 250 to 500 nm and the special absorption value at 396 nm is shown in Figure 8b. As can be seen from Figure 8, the AFs containing nano-SiO_2_ on the surface have the stronger absorbance than the untreated AFs, which illustrates that their capacity to absorb the UV light is related to the amount of nano-SiO_2_ decorated on the surface. Previous research into this phenomenon indicated that the introduction of silicon atoms could improve the UV absorption capacity [47,48]. Particularly, compared with the absorption value of the untreated fiber at 396 nm, the fibers treated under the pressure of 10, 12, and 14 MPa increases by 33%, 41%, and 52%, respectively, suggesting the improved absorption capacity of AFs with the increasing contents of nano-SiO_2_, which is in good agreement with the XPS results. 

#### 3.5.2. Mechanical Properties

Figure 9 gives the tensile strength, elongation to break, and tensile modulus of the AFs before and after 216 h of UV radiation. Figure 9a shows that the tensile modulus of the AFs decreases firstly and then increases with the number of nano-SiO_2_, contrary to the change for the elongation to break. The modified AFs exhibit considerably similar tensile strength as high as the untreated AFs, except for manifesting the increasing value of 2.61 GPa under the supercritical pressure at 14 MPa. Compared with the untreated AFs, the tensile strength, modulus, and elongation to break of the 14 MPa-modified AFs increase by 9.7%, 4.2%, and 6.4%, respectively, which may have contributed to the fact that some defects on the surface of fibers are repaired by the layered nanonetworks [42]. After exposure with 216 h of UV radiation, the tensile properties of fibers all decreased, as shown in Figure 9b, demonstrating that the UV light induced huge damage in the AFs. The AFs-UV shows a decrease in tensile strength, elongation to break, and tensile modulus, retaining only 73%, 91%, and 85% of the pristine AFs’ values, respectively, while those of AFs-SiO_2_-14MPa-UV are 91.5%, 98%, and 95.5%. The results can be attributed to the photostabilization of the nano-SiO_2_ particles on the surface, which absorbs the UV light and prevents degradation of the organic bonds [48]. Since the absorption becomes less and eventually disappears with the continuous consumption of nano-SiO_2_, the light will continue to destroy the amide bonds of AFs and lead to deterioration of its mechanical properties.

#### 3.5.3. Surface Compositions

Figure 10 shows the C_1s_ core-level spectra of AFs and irradiated AFs. The components of untreated AFs shown in Figure 10a are divided into three peaks at 284.2, 285.7 and 287.8 eV, corresponding to the C–C, C–N and C=O bonds, respectively. After the UV radiation, a new peak with respect to –COOH groups is observed at the binding energy of 289.8 eV for the irradiated untreated AFs shown in Figure 10b. Compared with the clear and strong –COOH peak of AFs-UV, the AFs-SiO_2_-UV sample shows relatively unobvious and weak peaks, which is due to absorption or reflection to UV light of the porous nano-SiO_2_, acting as a protective umbrella of AFs in the extent. Specifically, as shown in Figure 10c–e, the carboxyl peak in the nano SiO_2_-modified AFs treated under pressure from 10 to 14 MPa was gradually weaken, and almost invisible in the AFs-SiO_2_-12MPa-UV and AFs-SiO_2_-14MPa-UV samples, demonstrating that more effective protection is obtained in these samples due to their small size and rich contents in nano-SiO_2_ particles. Since the particle contents become less, the AFs-SiO_2_-16MPa-UV sample shows a large carboxyl peak shown in Figure 10f. These results indicate that nano-SiO_2_ can reduce the damage of UV irradiation to AFs.

#### 3.5.4. Surface Morphology

The surface morphologies of the untreated AF and irradiated AF shown in Figure 11a,b were investigated to further illustrate the changes in mechanical properties and surface compositions. As shown in Figure 11a, the pristine AF possesses a smooth and tidy surface attributed to its high crystallinity. However, after 216 h of UV irradiation, a relatively rough and cracked surface is observed in the AF-UV shown in Figure 11b, indicating that the irradiation causes cleavage of fiber chains and production of defects. Due to the occurrence of UV aging, the nanoparticles on the modified fibers disappeared. Moreover, it can be seen in Figure 11c–f that fewer cracks and stripes appear in the AFs treated under the pressures of 10 and 16 MPa; meanwhile, only a rough surface is found in the treatment at 12 and 14 MPa, implying that a smaller number of defects are observed in the SiO_2_-grown AFs. This phenomenon is attributed to the higher bond energy of Si–O–Si, suggesting that the nano-SiO_2_, to some extent, can reduce or even eliminate the influences of UV irradiation on AFs [49]. However, with the conversion of nano-SiO_2_ into other forms of energy after the absorption of UV light, the protective effects are weakened.

### 3.6. Interface Adhesion of Aramid Fiber/Epoxy

In order to evaluate the effect of nano-SiO_2_ on the bonding properties of AFs with resin, the interfacial adhesion of fiber/epoxy was characterized through the single fiber pull-out test. The interfacial shear stress (IFSS) of AFs with 120–190 µm droplets is calculated and presented in Figure 12a, and their average values are given in Figure 12b. It can be inferred that the IFSS of the modified fibers is significantly improved in terms of the distribution of data points. Compared with the IFSS of the untreated fibers, the fibers treated with nano-SiO_2_ under pressure of 10, 12, 14 and 16 MPa in Sc-CO_2_ increased by 27%, 50%, 64%, and 20%, respectively. These results can be attributed to the increased roughness of the fiber surface due to the introduction of nano-SiO_2_, thus enhancing the interfacial adhesion of AFs with epoxy resin. That means that the AF with a richer coating of nano-SiO_2_ has a higher IFSS. Growing nano-SiO_2_ on the surface of AFs is an effective way to improve its surface performance.

## 4. Conclusions

A facile strategy for directly growing nano-SiO_2_ on the surface of aramid fibers by supercritical carbon dioxide to improve thermal stability, IFSS, and UV resistance was reported. With different pressures, nano-SiO_2_ with different morphology, size, and quantity was observed on the fiber surface, demonstrating that the supercritical pressure can affect the size of nano-SiO_2_ particles and the greater the pressure, the smaller the size. The nano-SiO_2_-decorated fibers exhibited improvement in residual quality by 12% compared with the untreated fibers. Compared with the untreated fibers, the tensile strength, modulus, and elongation to break of the AFs-SiO_2_-14MPa sample correspondingly increased by 9.7%, 4.2%, and 6.4%, respectively. After exposure to 216 h of UV radiation, the AFs-UV sample showed decreases in tensile strength, elongation to break, and tensile modulus, retaining only 73%, 91%, and 85% of the values of pristine AFs, respectively, while the AFs-SiO_2_-14MPa-UV sample retained 91.5%, 98%, and 95.5%. The IFSS of the fibers treated with nano-SiO_2_ under 14 MPa pressure in Sc-CO_2_ increased by as much as 64%. Therefore, the nano-SiO_2_ particles greatly enhance the suitability of AFs in harsh environments and improve the interfacial adhesion with resin without destroying the tensile properties. In short, this study presents a green method for growing nano-SiO_2_ on the surface of AFs by Sc-CO_2_ to enhance the fiber′s thermal stability, IFSS, and UV resistance.

## Figures and Tables

**Figure 1 polymers-11-01397-f001:**
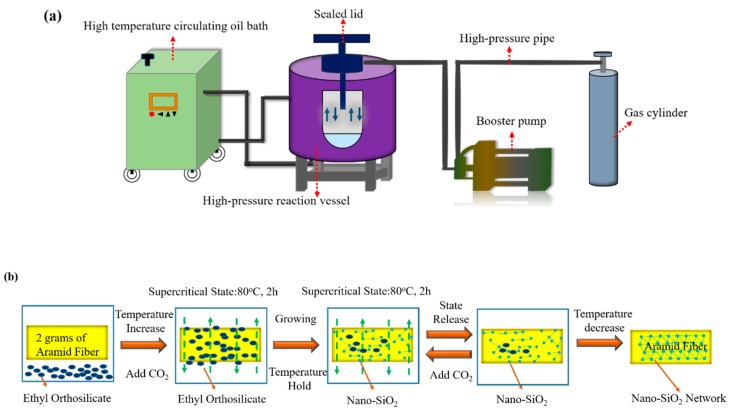
Set of reaction equipment (**a**) and schematic diagram of growing nano-SiO_2_ on the fiber surface (**b**).

**Figure 2 polymers-11-01397-f002:**
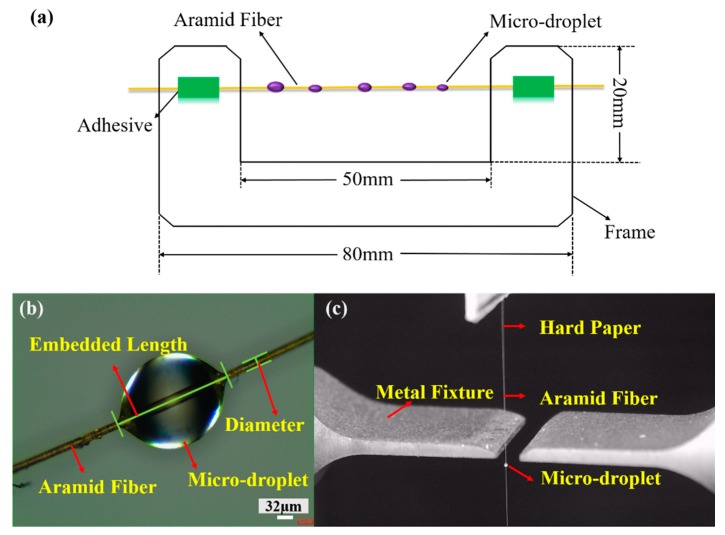
Simulation diagram of depositing microdroplet on the fiber (**a**). Images of AF–microdroplet (**b**) and the debonding device (**c**).

**Figure 3 polymers-11-01397-f003:**
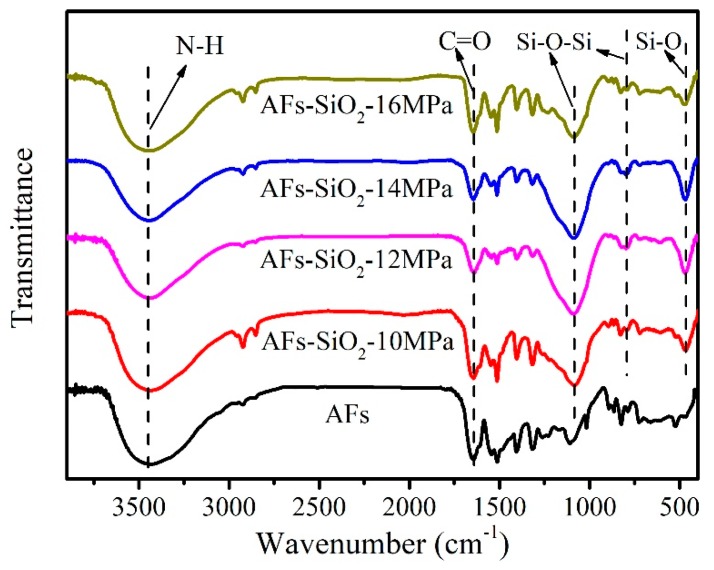
Infrared spectrum of the untreated and nano-SiO_2_-modified fibers.

**Figure 4 polymers-11-01397-f004:**
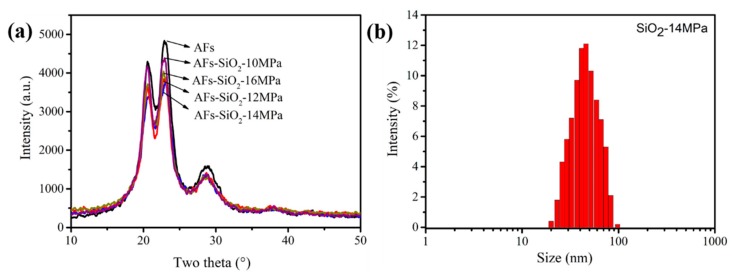
XRD patterns of the untreated and nano-SiO_2_ modified fibers (**a**). Particle size distributions of nano-SiO_2_-14MPa (**b**).

**Figure 5 polymers-11-01397-f005:**
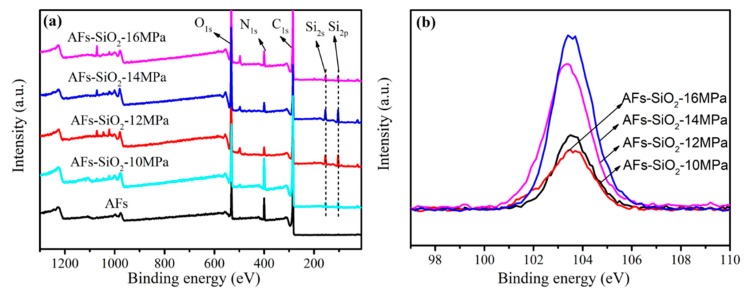
XPS wide scan of the untreated and SiO_2_-modified AFs (**a**); Si_2p_ core-level spectra of SiO_2_-modified AFs (**b**).

**Figure 6 polymers-11-01397-f006:**
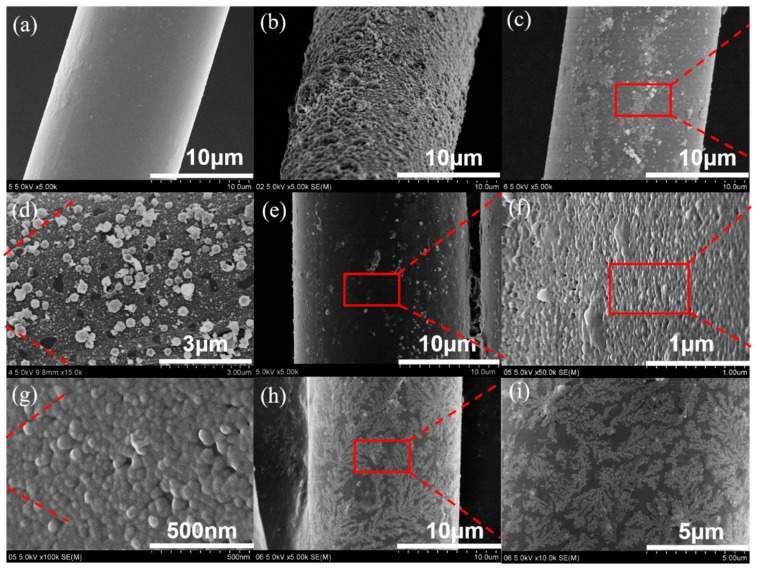
SEM images: AFs (**a**); AFs-SiO_2_-10MPa (**b**); AFs-SiO_2_-12MPa (**c, d**); AFs-SiO_2_-14MPa (**e–g**), and AFs-SiO_2_-16MPa (**h,i**).

**Figure 7 polymers-11-01397-f007:**
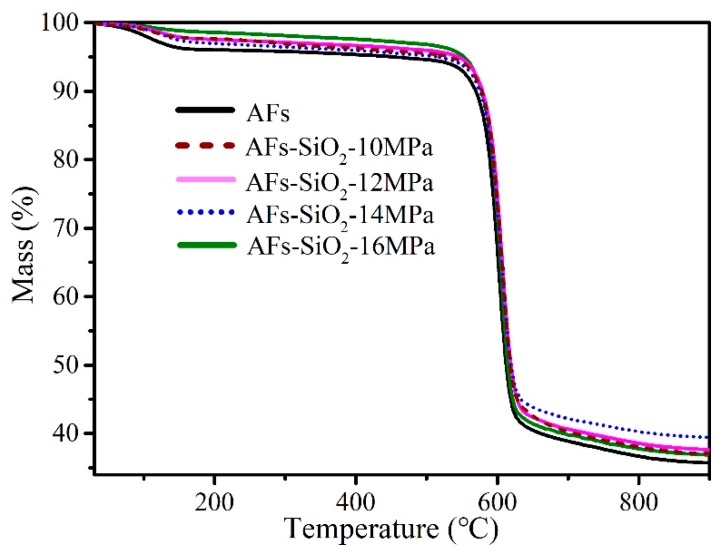
TGA curves of untreated fibers and modified fibers.

**Figure 8 polymers-11-01397-f008:**
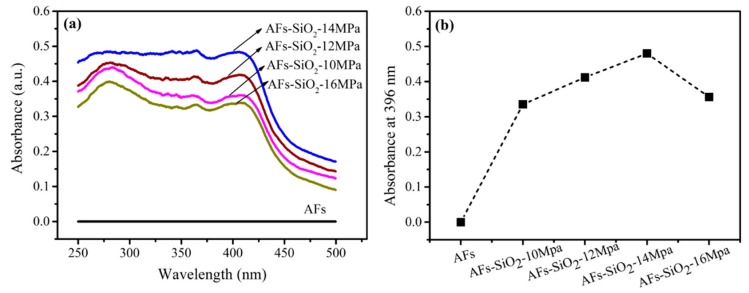
UV–Vis spectra of untreated and SiO_2_-treated AFs (**a**) and their absorbances at 396 nm (**b**).

**Figure 9 polymers-11-01397-f009:**
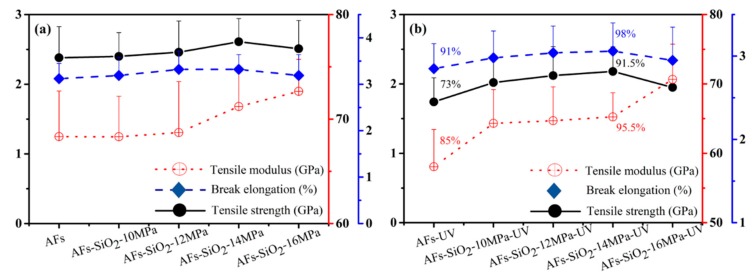
Mechanical properties of AFs and nano-SiO_2_-modified AFs before (**a**) and after (**b**) 216 h of UV radiation.

**Figure 10 polymers-11-01397-f010:**
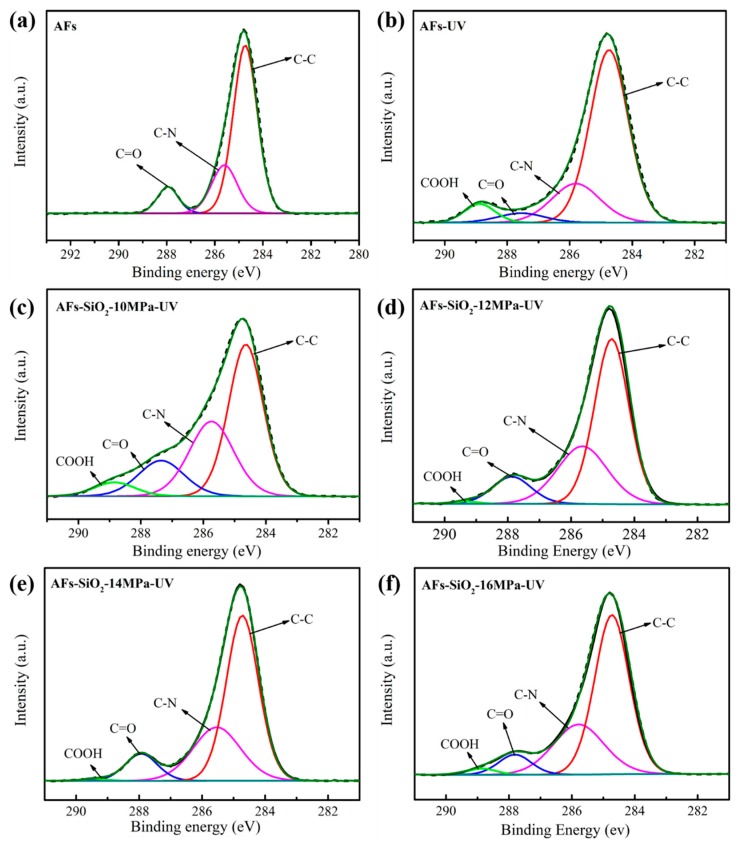
C1s core-level spectra of AFs and irradiated AFs.

**Figure 11 polymers-11-01397-f011:**
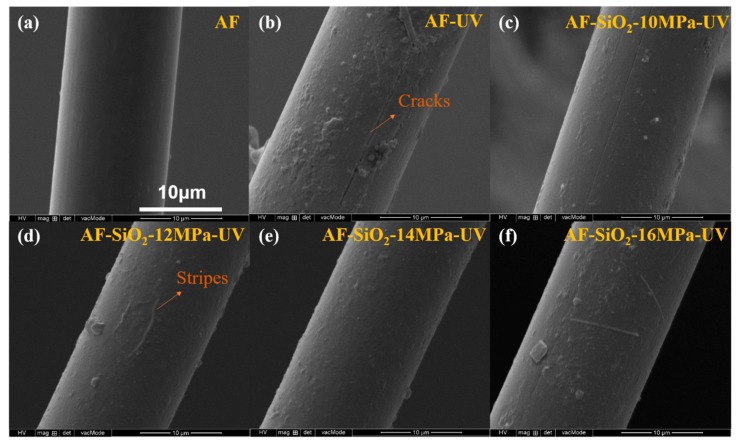
SEM images of the untreated and irradiated AF.

**Figure 12 polymers-11-01397-f012:**
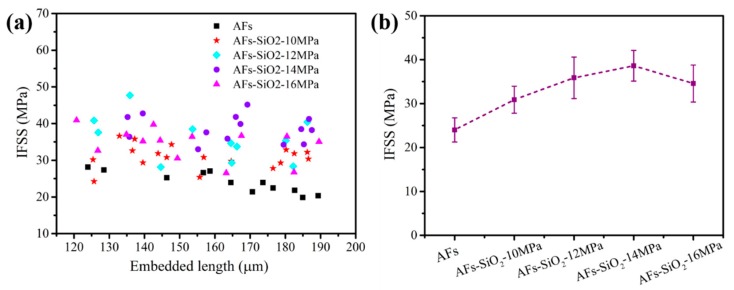
IFSS variation with different embedded lengths of the untreated and nano-SiO_2_ modified AF (**a**) and their values (**b**).

**Table 1 polymers-11-01397-t001:** Surface elements of AFs before and after treatment with nano-SiO_2_.

Sample	Atomic Percent (%)				Atomic Ratio	
	C	N	O	Si	O/C	Si/C
AFs	77.49	8.78	13.73	0	0.18	0
AFs-SiO_2_-10MPa	65.70	6.98	22.22	5.10	0.34	0.07
AFs-SiO_2_-12MPa	54.29	5.12	29.91	10.68	0.55	0.20
AFs-SiO_2_-14MPa	50.05	5.62	32.64	11.69	0.65	0.23
AFs-SiO_2_-16MPa	68.55	7.26	20.04	4.15	0.29	0.06

**Table 2 polymers-11-01397-t002:** Residual mass of the aramid fibers treated in Sc-CO_2._

Different Treatment Conditions	Residual Mass (%)	Nano-SiO_2_ Mass (%)
Untreated AF	35.70	0
AF-SiO_2_-10MPa	36.88	1.18
AF-SiO_2_-12MPa	37.62	1.92
AF-SiO_2_-14MPa	39.86	4.16
AF-SiO_2_-16MPa	37.12	1.42

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
