# Peer review of "Growing Nano-SiO2 on the Surface of Aramid Fibers Assisted by Supercritical CO2 to Enhance the Thermal Stability, Interfacial Shear Strength, and UV Resistance"

_polymers, 2019, doi:10.3390/polym11091397_

Round 1

Reviewer 1 Report

The manuscript reports on a method for nano-SiO2 coating of aramide fibers and contains extensive characterization of changes of fiber properties.  A special sol-Gel coating method using sc-CO2 as the main solvent was used in this study. In particular the mechanical properties of the fibers after artificial photoaging were show to improve considerably.

Aramid fibers are polymers of steadily increasing importance for high performance and outdoor applications. Therefore, the suggested coatings for performance improvement might by interesting for industrial applications and the materials were thoroughly characterized in the paper. However there are a few shortcommings in the data evaluation and interpretation, as detailed below, that should be revised before publication of the manuscript.

p. 5, last paragraph 167-168: The FT-IR peaks at 470 cm-1, together with the peaks at 1089 and 800 cm-1, are indicative of the presence of SiO2 coatings but I don’t see that any direct proof of real chemical bonding between the SiO2 and Aramide can be obtained from this IR bands.

p. 6, 174-177: The pronounced decrease in crystallinity with higher sc-CO2 pressure, shown by a steep decrease of diffraction peak intensities in Fig. 4, is not easily explained just by a simple increase of amorphous SiO2 coating thickness. This would mean that for the SiO2 coated fibers, a major part of the fiber consists of the SiO2 coating. Such thick coatings were not confirmed neither by increase of fiber diameter in SEM images, nor by XPS spectra, nor by residual mass in TGA experiments. An alternative explanation could be that TEOS precursor can diffuse to the interior of the aramide fibers in sc-CO2 and thus SiO2 particles are also formed in the interior bulk phase of the aramide fibers in contrast to the fiber surface SiO2 deposition discussed in the manuscript.

The build-up of intercalated  SiO2 nanoparticles in the aramide fiber bulk phase, are expected to strongly influence the crystallinity of the aramide fibers. The amount of SiO2 deposited on the fibers after different coating methods should be estimated from Fig.4 , in order to evaluate coatings. It should be compared to other results from XPS, TGA, etc. characterization of the fibers. The possibility of internal SiO2 formation inside the bulk aramide fibers should be discussed taking into account XPS. TGA, UV-Vis and XRD results

p. 7: Additionally, more complete information on the chemical structure of the used aramide fibers (para- or meta- aramide and chemical nature of the aryl-groups) should be reported. This would allow to assign e.g. XPS spectra to the theoretical atomic composition of the pristine aramide fibers taking into account complimentary data from FT-IR, XRD, TGA, etc.

p. 7, 198 ff. A more convincing explanation and detailed discussion should be given for the abrupt decrease in SiO2 coating density between 14 and 16 MPa sc-CO2 pressure. 

p.9: ch. 3.5.1., Fig. 8: Amorphous SiO2 phases generally show very high transparency (low absorbance) in the visible region reaching quite far into the UV region. Therefore, I suppose the reduced transparency at wavelength < 500 nm is mainly due to higher light scattering by SiO nanoparticles than by real absorbance

p. 9, Figure 7: from the residual mass at 850°C it should be possible to retrieve a rough estimate of the mass of SiO2 coatings for the different deposition conditions (pressures)

p. 10, 267-268: With respect to the applied aging by 216 h- UV radiation in the used wheathering test instrument, it should be reported what is the equivalent time under normal environmental conditions?

Further minor corrections and typos:

Abstract, p.1, 16-17:  The expression “UV resistance in the supercritical CO2” is misleading.  It should be written like “… a method to synthesize nano-SiO2 in supercritical CO2 on the AF surface to improve their UV resistance, thermal stab. ….”

p. 1, 32: what does “social production” mean here ?

p. 2, 43: “… advanced workers …” ???? probably means “advanced studies” or similar

p. 9, 245-246 “residual quality” probably means “residual quantity” ?

p. 10, Fig. 9: the visibility and legibility of the graphs should be improved by modifying the diagram scaling. E.g tensile modulus scale might be from 60-80 GPa, tensile strength from 2 to 4 GPa , etc,

Reviewer 2 Report

The article is well written but before publication I please add some information.
Figure 1 - I suggest to add some details in the Figure. The specific values of temperature, time, mass should be placed in the Figure description. It will made the paper easier to be read.

line 76 - the authors have written "The shapes and sizes of nanoparticles have been controlled for uniform growth on the aramid surface using supercritical technology" what does "supercritical technology" mean?

part 2.3 each method should be described in separate line

line 230 there is written: "Moreover, a network-like structure of nano-SiO2, like “honeycomb” shape, is observed on the surface of AFs treated at 14 MPa, which will protect the fibers from environmental
erosion" - please claryfy it.

Table 2 the value of Residual Mass (%) increases but at the preasure 16MPa decreases - how the authors can explain that phenomena.

Fig 8 - the samples prepared at 16 MPa can be characterized with smaller value of absorbance - can the authors explain why?
Fig 10b and line 317 - the comment is as above but the question is about IFSS

line303 there is written "It is obvious that fewer cracks and stripes appear in the AFs treated under the pressure at 10 MPa and 16 MPa' - it is obvious for the authors but not for each readers, so please add some explanations.
